# Bearing Capacity near Support Areas of Continuous Reinforced Concrete Beams and High Grillages

Zeljko Kos [1,*], Yevhenii Klymenko [2], Irina Karpiuk [3] and Iryna Grynyova [4]

1    Department of Civil Engineering, University North, 42000 Varazdin, Croatia
2    Department of Reinforced Concrete Structures and Transport Structures, Odessa State Academy of Civil Engineering and Architecture, Didrichson Street 4, 65029 Odessa, Ukraine; klimenkoew57@gmail.com
3    Department of Bases and Foundations, Odessa State Academy of Civil Engineering and Architecture, 65029 Odessa, Ukraine; irina.carpyuk@gmail.com
4    Department of Architecture Structures, Odessa State Academy of Civil Engineering and Architecture, 65029 Odessa, Ukraine; grynyova@ogasa.org.ua
*    Correspondence: zeljko.kos@unin.hr; Tel.: +385-98-757-989

**Abstract:** This work presents a proposed engineering method for calculating the bearing capacity of the supporting sections of continuous monolithic reinforced concrete tape beams, which combine pressed or driven reinforced concrete piles into a single foundation design. According to the mechanics of reinforced concrete, it is recommended to consider the grillage to be a continuous reinforced concrete beam, which, as a rule, collapses according to the punching scheme above the middle support (pile caps), with the possible formation of a plastic hinge above it. The justification for the proposed method included the results of experimental studies, comparisons of the experimental tensile shear force with the results of calculations according to the design standards of developed countries, and modeling of the stress-strain state of the continuous beam grillage in the extreme span and above the middle support-pile adverse transverse load in the form of concentrated forces. The work is important, as it reveals the physical essence of the phenomenon and significantly clarifies the physical model of the operation of inclined sections over the middle support. The authors assessed the influence of design factors in continuous research elements, and on the basis of this, the work of the investigated elements under a transverse load was simulated in the Lira-Sapr PC to clarify the stress-strain state and confirm the scheme of their destruction adopted in the physical model by the finite element method in nonlinear formulation. Based on the analysis and comparison of the experimental and simulation results, a design model was proposed for bearing capacity near the supporting sections of continuous reinforced concrete beams and high grillages that is capable of adequately determining their strength.

**Keywords:** reinforced concrete continuous beam; grillage; average support; stress-strain state; calculation

## 1. Introduction

The problem of the resistance of reinforced concrete structures to the combined actions of bending and torque moments along with transverse and longitudinal forces is one of the most important in the theory of calculating the properties of reinforced concrete. It has captured the attention of researchers for more than one hundred years, but it has not yet been fully resolved.

At the beginning of the twentieth century, the two main directions used in the calculation of concrete structures were formed. Some researchers extended theoretical solutions of the theory of elasticity into plastic, which led to the creation of the theory of plasticity. The basis of this trend was the rejection of the simple form of Hooke's law and its replacement with a complex dependence. The main goal of this direction was to predict the behavior of the structure at all stages of the load, from the initial state to the ultimate state (i.e., to

destruction). In this direction, the most famous works are those of Nadai, Prager, Mises, Genka, Ilyushin, and others.

The second way of calculating normal sections is based on considering only the limiting state without considering the previous load history, which allows the determination of the breaking load for the new design scheme that the structure acquires in the limiting state. The discoverer of the second direction was A.F. Lolate, who, in 1905, proposed considering the instantaneous equilibrium of ultimate forces before destruction. This was later approved in 1932 and recommended for use in design practice and, in 1938, was reflected in the "norms and technical conditions" (NiTU-38) with a curved diagram of the stresses of compressed concrete. In 1944, at the suggestion of P. L. Pasternak, this diagram was replaced by a rectangular one. Invaluable contributions to the development of the theory of calculation of reinforced concrete structures of that period include the fundamental works of A.A. Gozdeva, M.S. Borishansky, Y.V. Stolyarova, and V.I. Murasheva et al.

In recent decades, the deformation method for calculating normal sections of reinforced concrete structures [1], which can be attributed to the third stage of its development, has been actively distributed, incorporated into the regulatory documents of most European countries, and borrowed at one time in the USSR. In recent years, the deformation-force model [2] has undergone significant development. It allows the calculation of building structures for groups I and II of limiting states from a single perspective.

The question of the possible transformation of a reinforced concrete element after the formation of inclined cracks into a spacer system was first posed in 1909 by A.N Talbot. E. Mörsch's idea regarding the perception (by clamps) of the main tensile stresses led to a design scheme for a reinforced concrete beam in the form of a truss or an arch. In the first case, the role of the upper compressed zone is played by the concrete of the compressed zone, and the longitudinal reinforcement serves as the stretched zone. The zones are connected through a lattice in which the stretched elements are represented by transverse reinforcement and the compressed elements are represented by imaginary concrete braces at an angle of 45° to the longitudinal axis of the beam.

Further research in this direction was aimed at improving the E. Mörsch truss analogy. In particular, he suggested that 20% of the transverse force with a concentrated load (and 40% with a uniformly distributed load) be transmitted to the longitudinal reinforcement and the stretched part of the concrete. However, this proposal has not received theoretical confirmation.

F. Leonhardt contributed to the development of this method [3,4]. He found that the calculated angle of the inclined crack $\alpha$ did not correspond to the experimental results, concluding that it was necessary to take into account not only the equilibrium conditions but also the deformability of the materials, the compatibility of the deformations of rigid concrete elements, and the compliance of the steel rods of the transverse reinforcement. He discovered the dependence of the angle $\alpha$ on the ratio of the width of the compressed shelf to the thickness of the ribs for T-sections and pointed out the need to consider the indented effect.

In the method of modified truss analogy, P. Pegan [5] et al., in contrast to the previous method, considered concrete in the compressed zone to be a part of the external shear force.

T. Zsuttu proposed a phenomenological method [6] based on experimental data and methods of mathematical statistics and included empirical expressions, by which the author tried to describe the physical picture of the phenomenon for a relatively narrow range of problems.

V.I. Virschilas, A.D. Shpukshta, and A.P. Kudzis developed a phenomenological method [7] based on a statistical evaluation of the strength and crack resistance of inclined sections of reinforced concrete elements. The main drawback of this approach was the lack of a clear physical picture of the processes occurring in the supporting areas and the difficulty of transferring them to other elements.

In the USA and Canada, the critical oblique crack method has achieved dominance. This method is based on the hypothesis that the transverse force acting on an element is

perceived by transverse reinforcement (in which extreme stresses arise) and concrete (the stress in which corresponds to the stress of formation of such cracks).

The entire course of the further development of the theory of reinforced concrete showed that these methods, considering their conventions, have not adequately reflected the real work of the supporting sections of reinforced concrete elements.

To eliminate the shortcomings of the classical theories of the 1930s and 1940s, domestic scientists carried out extensive experimental studies to create a method for calculating reinforced concrete structures. They focused not only on bending and torque but also on transverse and longitudinal forces in their calculation methods for destructive efforts. One of the first attempts to develop such a method was the work of V.I. Murasheva. At the same time, under the guidance of A.A. Gvozdeva, M.S. Borishansky created a new method for calculating the inclined section on the action of a transverse force in the fracture stage [8]. In this method, efforts are perceived only by concrete over an inclined crack and transverse reinforcement; it intersects it.

According to A.S. Zalesov and Yu. A. Klimov [9], the greatest prospects, from the point of view of constructing and improving engineering methods of calculation around the support sections of beams, were found in the methods of A.A. Gvozdev and M.S. Boribshansky.

The method of limiting equilibrium in an inclined section has undergone great changes, as amended by SNiP2.03.01-84* [10], in which it was recommended to determine the bearing capacity of a bearing section, not only by an inclined crack under the overwhelming effect of a transverse force or bending moment, but also by an inclined strip between inclined cracks in the direction of the main compressive stresses. These changes also concern the expression for the transverse force as perceived by concrete (two additional empirical coefficients, $\varphi_n$ and $\varphi_f$, were additionally introduced, and the coefficient $\varphi_{b2}$ was introduced instead of the coefficient $k_2$), the calculation scheme, and the construction of the calculation itself, for determination of the length of a dangerous inclined crack.

However, according to the authors of [9], one of the reasons for the unsatisfactory convergence of the shear forces calculated according to [10] and the experimental values was the lack of such internal forces as the concrete adhesion forces in the critical inclined crack and the tensile forces in the longitudinal reinforcement in the calculation model. It should also be noted that the method in [10] did not take into account several other factors that have a significant effect on the bearing capacity of inclined sections of reinforced concrete structures. These include the presence of longitudinal tensile forces applied with an eccentricity, changes in the length and height of the dimensions of the cross-section, the combined influence of the transverse force and other force factors, the technological, mechanical and power damage of concrete, the nature and mode of action of the external load, and temperature effects, among others.

In addition to generally accepted calculation methods, other methods were developed in parallel with them in the same period.

A significant achievement in the theory of reinforced concrete is the method of A.S. Zalesov and Yu. A. Klimova [9]. They designed a physical model of a reinforced concrete beam that gradually turns from a solid body into a disk-bond system. The destruction of this system begins with the exclusion of one of the connections from the work, the crushing of the concrete of the compressed zone over the top of the critical inclined crack, the flow of longitudinal reinforcement at the place of its intersection by an inclined crack, a slice of concrete over an inclined crack with small spans of a slice with a small amount of transverse reinforcement, or destruction along an inclined compressed strip between inclined cracks. However, the practical application of this method has encountered some problems (e.g., the determination of six correlated empirical coefficients, $\omega_i$, using two iterative processes, etc.).

Additionally, the calculation method of L.A. Doroshkevich, B. Demchina, and S.B. Maksimovich [11] is of interest. The authors related the calculation of the strength of inclined and normal sections based on the known differential dependence of bending theory ($dM/dx = Q$), taking the angle of inclination of the calculated section of the longitudinal

axis of the element to 45°. However, the inconsistency of the change in tensile force in the longitudinal reinforcement and the diagram of bending moments was taken into account by the doubtful coefficient of the diagram of moments, *V*.

The deformation calculation method [12] of the supporting sections of the span-reinforced concrete structure proposed by A. Davydenko and others is noteworthy. In it, the destructive shear force and other parameters of its bearing capacity are determined through the strength of normal sections using diagrams of the state of materials and other hypotheses inherent in this method. However, the schemes and models are still far from perfect. They apply to the selected criteria for the strength of reinforced concrete, considering its complex stress-strain state under the action of various force factors and external factors, the form of fracture, the projection size of a dangerous inclined crack, the nagging effect of longitudinal reinforcement, and other factors.

The deformation model [13,14] of reinforced concrete structures under the action of bending moments and longitudinal forces, proposed by A.I. Zvezdov, A.S. Zalesov, T.A. Mukhamedievim, and E.A. Chistyakov, significantly improved the new regulatory documents of Russia. At the same time, the analysis performed by the authors [13,14] showed that numerous new developments in the strength of inclined sections of reinforced concrete structures had not yet reached the level at which they could be accepted as normative methods [15]. Therefore, in [15], a simplified model for calculating plane and spatial oblique cross-sections was adopted in comparison with [10].

Summarizing the above, it can be emphasized that the described studies on the study of strength near the support sections of continuous reinforced concrete beams and grillages using various authors' methods are not generalized and are not able to give a final answer to the question of the general influence of the main design factors and external factors on the nature of deformation, cracking, and destruction of these sections of spanning reinforced concrete structures. In particular, the redistribution of internal forces in these areas before the destruction of these elements, the magnitude of the indentation effect of longitudinal reinforcement, the adhesion forces along the banks of a dangerous inclined crack, the dependence of the shape or pattern of their destruction on the ratio of structural factors and external factors, and other similar factors are not well understood.

Further developments of the theory of strength of inclined sections of bar-reinforced concrete structures and the search for appropriate design models are currently being addressed in three main areas:

- The accumulation of new experimental data and accounting or improvement of the empirical dependencies of the method of M.S. Borishansky [8];
- An in-depth study of the nature of the deformation, cracking, and fracture of reinforced concrete elements in areas of joint action of tough and large moments, transverse and longitudinal forces [9,16], the development of an analytical apparatus, the development an engineering methodology for calculating the limit state [12], and others;
- The development and improvement of the deformation method and the deformation-force method for calculating span-reinforced concrete structures [17,18], which allows one to predict the deformability, crack resistance, and strength of a reinforced concrete element throughout its volume, to determine the dangers of normal, flat, or spatial inclined sections in the general case of stress—that is, with the simultaneous action of transverse and longitudinal forces, bending and torque, and other influences.

The second direction is most promising for further improvements of engineering methods for calculating these structures, whereas the development of the science of reinforced concrete in general is the focus of the third area.

An analysis of recent publications [19–27] on this issue showed that they have mainly dealt with the strength of normal sections and the redistribution of internal forces within them. At the same time, the strength of the inclined sections of continuous beams has been the subject of considerably less focus [28–31]. At the same time, there is no consensus on an exhaustive answer for making a reliable forecast of the durability for health.

In [32], the influence of high temperature curing and surface humidity on the tensile strength was studied; in our study we considered normal temperature and humidity. Article [33] also studied the effect of the composition of polyethylene composites on structural repairing of materials.

Based on the above, the purpose of this article was to develop and substantiate an engineering method for calculating the bearing capacity of most loaded continuous reinforced concrete beams and high grillages near support sections.

*Research Objectives*

- Select, systematize, and analyze the data of previously conducted experimental studies [3,34];
- Assess the influence of design factors in continuous research elements on their bearing capacity;
- Develop a physical model of the work and destruction of these structures under transverse load;
- Assess the possibility of using the calculation formulas of national design standards of developed countries to predict the bearing capacity of these areas by comparing the results of calculations with experimental data;
- Simulate the work of research elements under transverse load to clarify the stress-strain state and confirm the scheme of their destruction, adopted in the physical model by the finite element method in a nonlinear formulation;
- Propose a design model of the bearing capacity near the supporting sections of continuous reinforced concrete beams and high grillages that is capable of adequately determining their strength.

## 2. Materials and Methods

In this regard, the Odessa State Academy of Civil Engineering and Architecture conducted systematic experimental studies [16,34,35] of the bearing capacity around the support sections of complexly loaded reinforced concrete structures, including continuous high beams. These structures symbolize the work, including monolithic reinforced concrete high grillages resting on the heads of piles, and from above, they are unfavorably loaded with concentrated forces (for example, columns). This work presents an engineering method for calculating the bearing capacity of the supporting sections of continuous monolithic reinforced concrete tape beams based on a punching scheme and includes the results of experimental studies from [16,34].

To achieve this goal, a series of in-situ experiments with two-span continuous reinforced concrete beams were implemented on the relevant state budget topics.

An analysis of the literature showed that researchers have not yet developed a unified idea; there are various interpretations about the influence of structural factors and external factors on the bearing capacity of the supporting sections of continuous reinforced concrete structures.

From literary sources, it is known that the main performance parameters of reinforced concrete structures obey normal Gaussian distribution law.

Because research factors can influence the exit function nonlinearly, it was advisable to approximate it by a polynomial of the second degree. Therefore, the experimental samples of this series (*V*) of experiments were made according to a five-factor three-level, similar in properties to the D-optimal, Ha5 plan [36]. This plan provided the same forecast accuracy of the output parameter in the region, which is described by a radius equal to 1 relative to the zero point.

The following factors were selected as research and changed at three levels:

$x_1$—is the relative run of the slice, $a/h_0 = 1, 2, 3$ with $h_0 = d = 155$ mm;

$x_2$—class of concrete C, MPa, C12/15, C20/25, C30/35;

$x_3$—coefficient of transverse reinforcement $\rho_w (B_p I) = 0.0018; 0.0032\ 0.0050$ ($2\emptyset3, 4, 5\ B_p I$, S = 77,5 mm);

$x_4$—coefficient of lower longitudinal reinforcement $\rho_{l,b}$(A500 C) = 0.0101; 0.0146; 0.0199 (2Ø10, 12, 14 A500 C);

$x_5$—coefficient of supreme longitudinal reinforcement $\rho_{l,t}$(A500 C) = 0.0101; 0.0146; 0.0199 (2Ø10, 12, 14 A500 C).

Each experiment within the full-scale experiment was provided with two specimen beams with six supporting sections. A total of 54 primary and 3 auxiliary seminatural beams were tested. The spans of the samples were 8 $h_0$ + 8 $h_0$ = 1240 mm + 1240 mm. They were reinforced (Figure 1) with two flat-welded frames. For the manufacture of the above beams, heavy concrete of the above classes was used on granite-crushed stone of fractions 5–10 mm and quartz sand with a particle size of 1.5. As a binder, ordinary 400 Portland cement (without additives) was used. To reduce the water-to-cement ratio, improve the easy-to-find concrete mix, and shorten the time required for concrete to gain strength in all experiments, we used the complex additive Relaxax—Super M (ISO 9001 No. 04.156.26) in the amount of 1% of the cement weight in terms of dry substance.

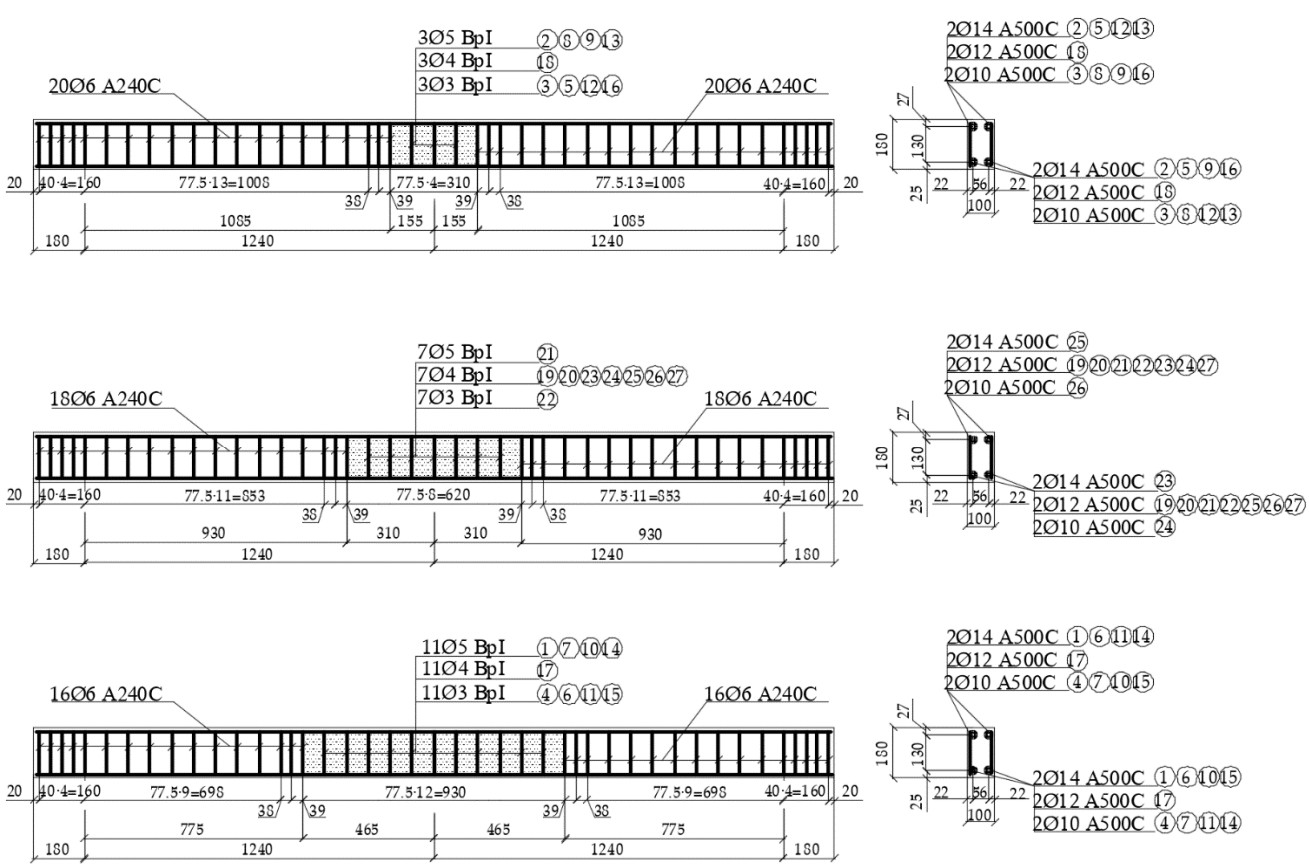

**Figure 1.** Design and reinforcement of continuous reinforced concrete beams (*V* series). *I*—experience number.

To test prototype beams, special power plants were designed and manufactured (Figure 2). Following the recommendations of regulatory documents, the prototypes were loaded with a hydraulic jack DG-50 and a distribution beam—traverse with two steps concentrated by short-term forces—until the first normal and inclined cracks appeared along (0.04 . . . 0.6)$F_{ult}$, then by (0.08 . . . 0.120)$F_{ult}$ to the development of maximum permissible deflections, and finally, by (0.04 . . . 0.06)$F_{ult}$ to destruction. The load exposure at each stage was 15 min, with all measurements at the beginning and the end of each degree of load.

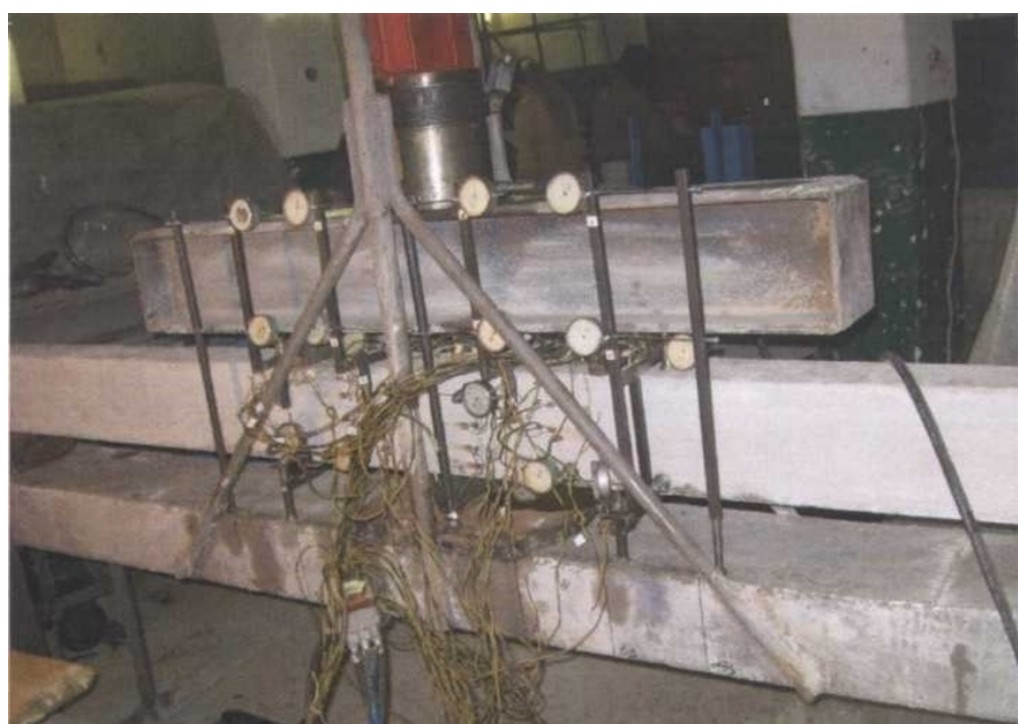

**Figure 2.** General view of the test of a continuous double-span beam.

Before the manufacture of the experimental beams, chains of strain gauges KF5P1-5-200 (with a base of 5 mm) were glued to the longitudinal compressed and stretched reinforcements of one of the flat frames. They were pasted per the technology recommended by the manufacturer («Veda» LLC, Kyiv, Ukraine).

The concrete deformations of the experimental samples were measured using strain gauges with a base of 40 and 50 mm, glued, according to the generally accepted technique, to one side and top polished faces. The transition from the stress-strain measurements in the experiment was carried out using Hooke's law, and in concrete, according to the secant modulus of elasticity. The deformation of concrete of the compressed zone and the tensile reinforcement was monitored using dial gauges I-1 ... 8 installed with a base of 100, 150, and 200 mm (Figure 3). Vertical movements were determined using dial gauges (P-1 ... 5). The tilt (rotation) angles of the middle support part, the support part, and the beam span were determined using dial gauges (B-1 ... 8).

Modeling of the stress-strain state of the research elements, in general, was carried out considering the proportionally increasing load by the transverse forces and the bending moment, utilizing a nonlinear finite element calculation in the Lira-Sapr PC using real phase diagrams of materials and the phenomenological strength criterion of G.A. Geniev, V.M. Kissyuk, and G.A. Tyupin.

Given the symmetry of the research elements, these calculations were performed for only one-half of the beam. It was conventionally divided into eight eight-node isoparametric volumetric elements (No. 236) with dimensions of 10 mm × 10 mm × 10 mm for convenience in modeling reinforcement, and also because granite-crushed stone of fractions 5 ... 10 mm was used as a large aggregate.

In the calculations, we used the step and step-iterative methods, using the step-iterative dependence No. 14 of the library with the corresponding algorithm.

Deformation, cracking, and fracture of research double-span-reinforced concrete beams occurred according to the rules of structural mechanics and were predictable (Figure 4).

Figure 5 shows photos from the in-situ experiments, which were implemented at the Odessa State Academy of Civil Engineering and Architecture.

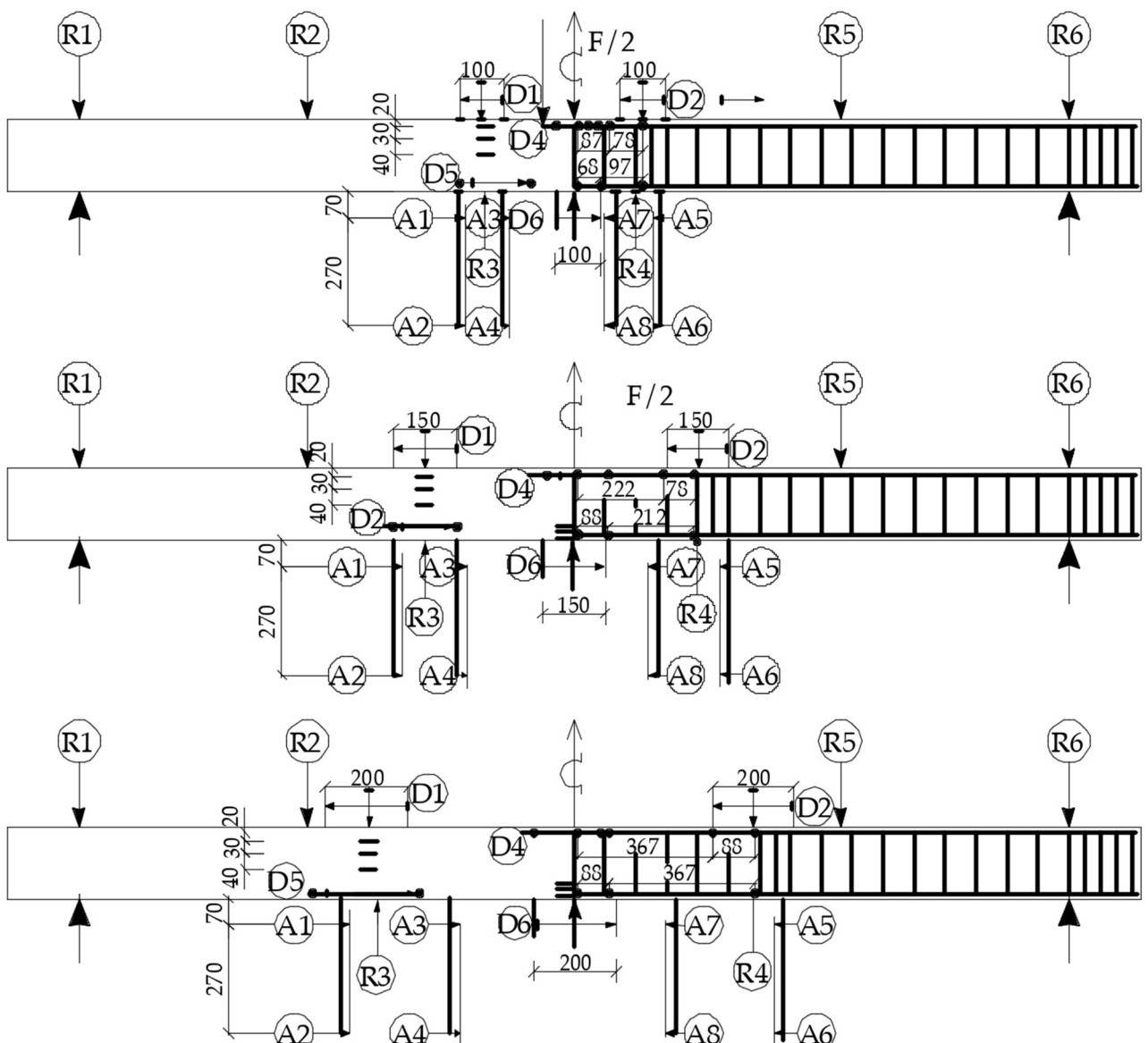

**Figure 3.** Scheme of loading, placement of devices and stickers of strain gauges in experimental two-span beams with small, medium, and large spans of the cut, where Ri—run-meter, Di—clock-type indicator for measuring concrete deformations in a compressed zone, as at the level of the center of gravity of a tensile reinforcement, Ai—clock-type indicator for the angles of rotation of normal sections of a continuous beam.

Normal cracks appeared in the zone of maximum bending moments above the average support. With a further increase in the transverse load, normal cracks above the middle support developed deeper into the beam, their opening width increased, and new normal flaws appeared in the spans under concentrated forces. Subsequently, the first inclined cracks appeared, and over the middle support, it became possible to form a plastic hinge with a redistribution of internal forces and an increase in flying moments with a constant value of the reference moment. A further increase in the transverse load led to the development of normal and inclined cracks with the predominant opening of inclined cracks to failure along inclined sections. Practically all experimental specimen beams of the expressed (**V**) series were destroyed along both inclined sections in the form of an inverted trapezoid according to the bursting pattern above the middle support, which was confirmed by the photo fixation given in [34].

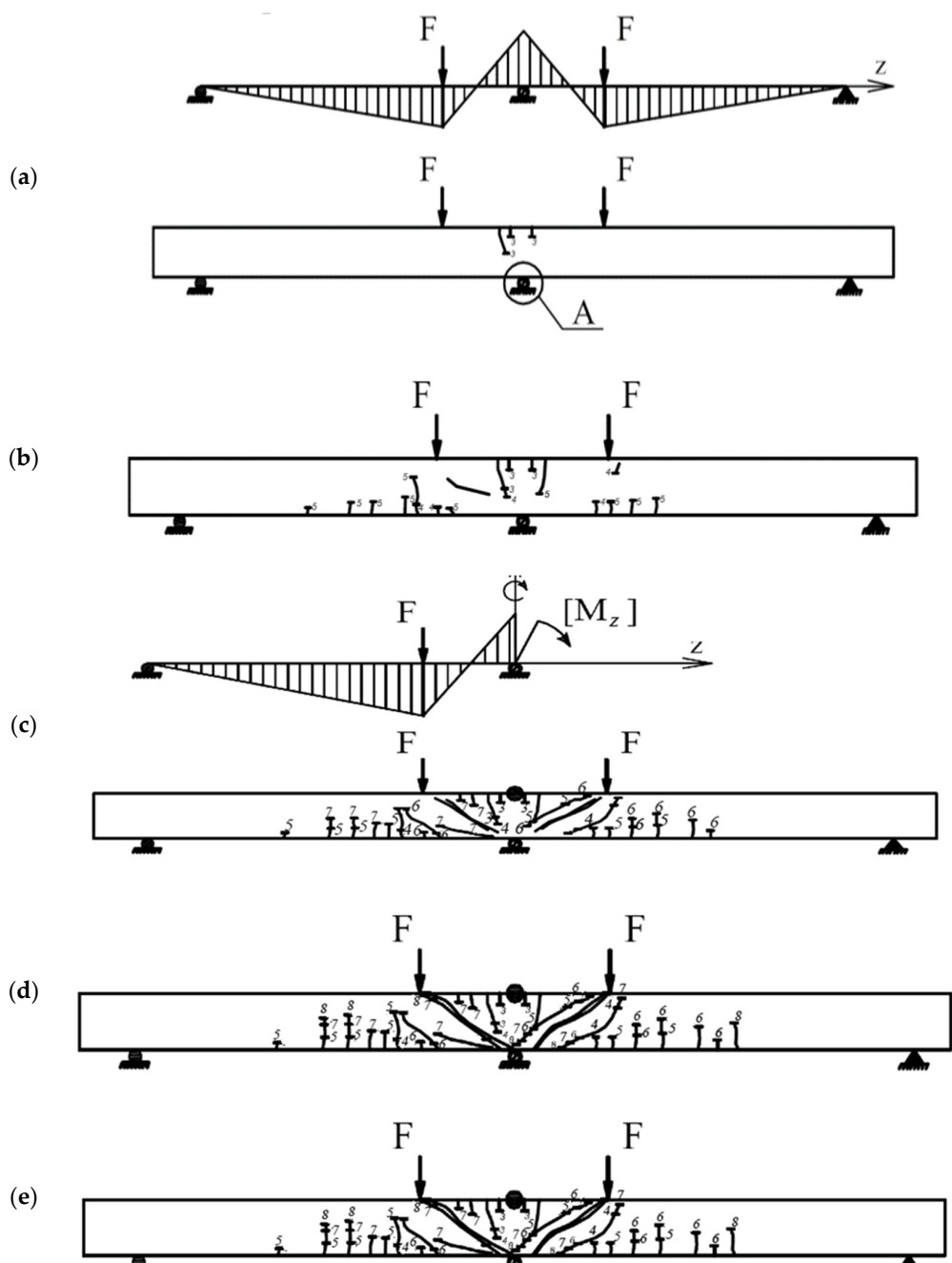

**Figure 4.** The mechanism of cracking, deformation, and fracture of a continuous two-span-reinforced concrete beam—grillage. Plot of $M_z$ before the appearance of the plastic hinge (**a**,**b**). Diagram $M_z$ after the appearance of a plastic hinge over the middle support (**c**–**e**). (**a**) The appearance of normal cracks above the middle support. (**b**) The development of normal cracks above the middle support, the appearance of normal cracks under concentrated forces (or inclined in the span of the shear). (**c**) The development of normal and inclined cracks, the appearance of a plastic hinge above the middle support, an increase in the flyby moment. (**d**) The development of normal and inclined cracks with a predominant opening of inclined cracks, up to failure along inclined cracks. (**e**) Fracture along inclined cracks with the formation of plastic joints under concentrated forces in beams with large and medium shear spans.

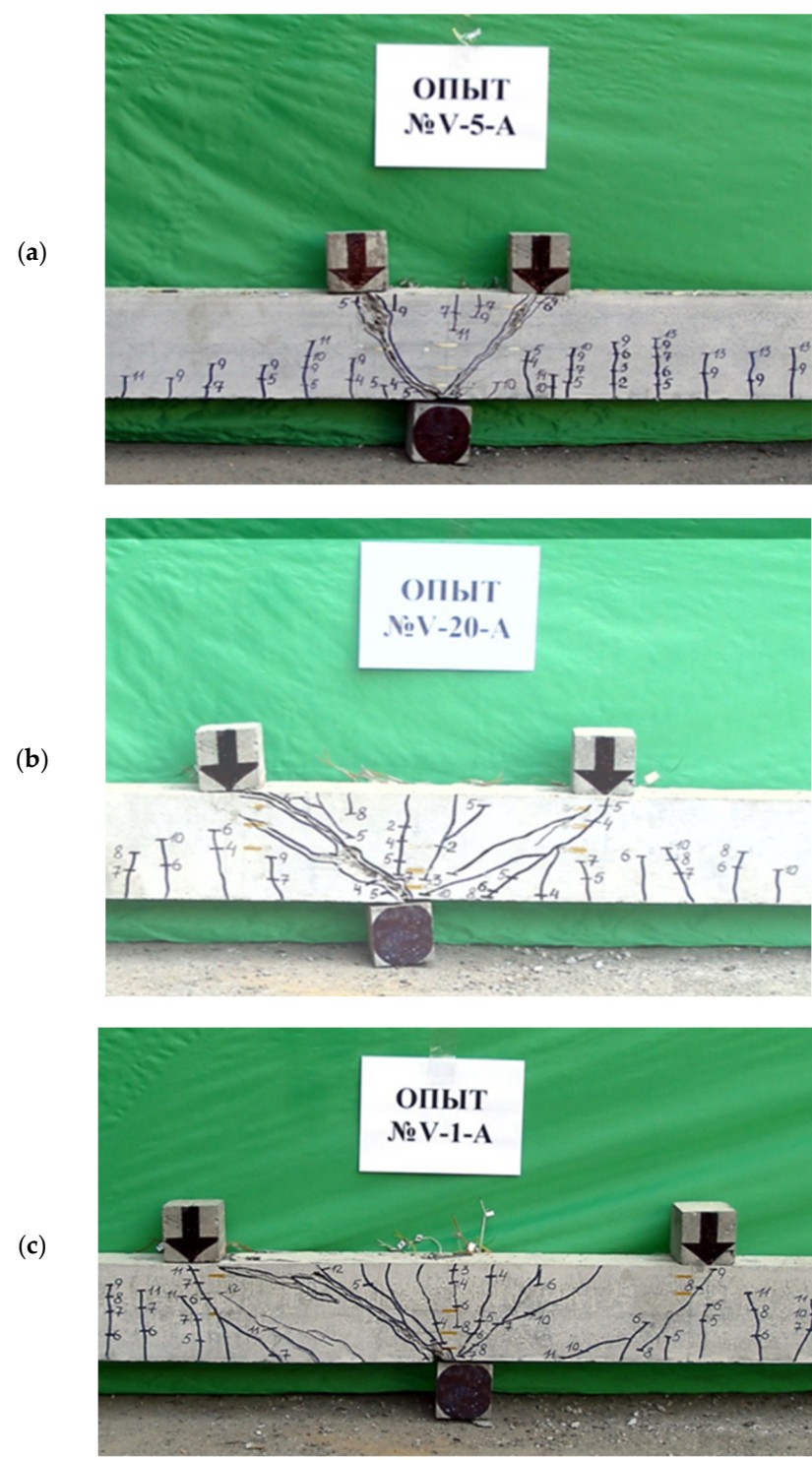

**Figure 5.** The nature of cracking and fracture of prototype beams with (**a**) small, (**b**) medium, (**c**) large cut-off spans.

The experimental specimen beams were designed with almost equal strength along normal and inclined sections, but such that their destruction occurred, nevertheless, along inclined sections when destructive transverse forces and related bending moments were exposed at the final stage of their operation.

As a result of processing the obtained experimental data, extracting insignificant data and recalculating those coefficients that remained using the effective COMPEX computer program, we obtained adequate experimental statistical dependences of the main working

parameters of the experimental samples. They showed good informational usefulness and good convergence with the experimental data (coefficients of variation did not exceed 8%).

### 2.1. The Strength of the Experimental Samples

$$Y[V_u/(bh_o)] = 5.10 - 1.98x_1 + 0.78x_2 + 0.06x_3 + 0.63x_4 + 1.01x_5 + 1.00x_1{}^2$$
$$-0.42x_1x_2-0.52x_1x_5 + 0.12x_2x_4 + 0.31x_2x_5, MPa, \tag{1}$$

where: $V_u$—breaking shear force with a coefficient of variation $v$ = 5.7% in a full-scale experiment; $b$, $h_0$—accordingly, the width and working height of the cross-section, which was introduced to enable comparison of the obtained data with the results of studies of other authors, as well as leveling the influence of the size of the cross-section (that is, a scale factor).

The presented adequate dependence (1) has a significant advantage over other statistical methods for setting up experiments and processing their results. The experimental-statistical dependence consists primarily of the fact that it allows one to evaluate the influence of each structural factor presented above on the strength of inclined sections, not only in particular, but also in interaction with each other.

Therefore, the bearing capacity of the bearing sections of experimental samples reduced to the working cross-sectional area increased to its average value of 5.10 MPa:

- With a decrease in the relative span of the slice, factor $x_1 = a/h_0$, from 3 to 1 by 78%;
- With an increase in the class of concrete (factor $x_2$) from B12/15 to C30/35 by 31%;
- With an increase in the coefficient of transverse reinforcement $\rho_w$ from 0.0018 to 0.0050 by 2.4%;
- With an increase in the number of lower longitudinal reinforcement $\rho_{1.b}$ from 0.0101 to 0.0199 by 25%;
- With an increase in the coefficient of the upper longitudinal (assembly) reinforcement $\rho_{1.t}$ from 0.0101 to 0.0199 by 40%.

All factors substantially interacted with each other. Destructive transverse force increased while reducing the span of the slice and increasing the class of concrete by about 8%. There was a reduction of shear span and increase of transverse reinforcement coefficient by 10%, an increase in the class of concrete and the number of lower longitudinal reinforcement by 2.4%, and an increase in the class of concrete and the coefficient of upper longitudinal reinforcement by 6%.

The presence of a quadratic effect of +1.00 $x_1{}^2$ indicated that a further increase in the destructive transverse force $V_u$ beyond the change in factor $x_1 < 1$ was of a damped nature with the possible formation of an extremum.

### 2.2. Deflections of Research Reinforced Concrete Beams

The maximum deflections in the continuous research elements at the initial stages of their operation and the operational level of the load were observed in the middle of the spans, and with further (load) increase in the deflections, they moved toward the action line of concentrated forces at the middle support.

The deflection ($f_v{}^{0.95Fu}/l$), reduced to the distance between the supports (l) (before the destruction (0.95 $F_u$) of the experimental beams) can be represented by the following relationship:

$$\hat{Y}\left(f_V^{0.95F_u}/l\right) = \left(2.44 + 0.33x_1 - 0.07x_4 + 0.31x_1^2 - 0.15x_5^2 + 0.06x_2x_3 + 0.12x_4x_5\right)10^{-3}, \qquad v = 6.7\%. \tag{2}$$

The influence of research factors on the value of deflections of experimental samples is adequate and can be described similarly to dependence (1).

### 2.3. Crack Resistance of Experimental Samples

The crack resistance of normal sections above the support and in the spans can be characterized using the bending moments given in the working area $(bh_0)$ and the characteristic tensile strength of concrete $(f_{ctk})$, respectively, according to the following mathematical models:

$$\hat{Y}\left[M_{cr.\perp.\text{sup}}/(f_{ctk}bh_0)\right] = (18.6 + 4.4x_2 + 0.8x_5)\,10^{-2}, m, \ v = 5.0\%; \tag{3}$$

$$\hat{Y}\left[M_{cr.\perp.\text{span}}/(f_{ctk}bh_0)\right] = (18.4 + 4.4x_2 + 0.8x_4)\,10^{-2}, m, \ v = 5.5\%. \tag{4}$$

The greatest among the design factors that influenced the relative moment of formation of normal cracks in continuous beams had a concrete class value (48%). Longitudinal tensile reinforcement values (9%) also coincided with the data of other researchers [3,9,37].

After the formation of normal cracks over the middle support and in the spans, with an increase in the transverse load, inclined cracks appeared on both sides of the average support on the lateral faces of the support sections of the experimental elements. They appeared approximately in the middle of the height of the beams running from the support to the places where external concentrated forces were applied.

The transverse force $(V_{cr})$ reduced to $(bh_0)$ and $(f_{ctk})$, which caused the appearance of the first inclined cracks in the inclined sections of the research elements, and can be characterized by experimental-statistical dependence (5):

$$\hat{Y}\left[V_{cr,/}/(f_{ctk}bh_0)\right] = 1.02 - 0.09x_1 + 0.23x_2 + 0.02x_1^2 - 0.02x_1x_2, v = 7.0\%, \tag{5}$$

which shows that the greatest positive impact on this output parameter has a concrete class of 45%.

### 2.4. The Width of the Opening of Normal and Inclined Cracks in the Experimental Elements

At the operational load level $(0.65F_u)$, the width of the opening of normal cracks above the middle support can be characterized by the following adequate model:

$$\hat{Y}\left(W_{k.\perp.\text{sup}}^{0.65F_u}\right) = 0.19 + 0.01x_2 + 0.04x_4 - 0.04x_5 + 0.03x_2x_4 - 0.02x_2x_5 - 0.01x_4x_5, \text{mm}, \ v = 8\%. \tag{6}$$

With a similar level of transverse load of the test samples, the width of the opening of inclined cracks, as a rule, did not exceed 0.3 mm. Prior to the destruction of their supporting sections, the following dependence can be characterized:

$$\hat{Y}\left(W_{k/}^{0.95F_u}\right) = 0.67 + 0.12x_1 - 0.04x_2 - 0.06x_3 - 0.04x_4 - 0.04x_5 - 0.06x_1x_2 + 0.11x_4x_5, \text{mm}, \ v = 7.8\%. \tag{7}$$

The average width of the opening of inclined cracks (0.67 mm) was fixed before the fracture of the samples was almost double the permissible values, primarily because they were designed so that their fracture occurred at the supporting sections while striving to maintain the principle of equal strength of normal and inclined sections of continuous beams.

### 2.5. Projection Length of Dangerous Inclined Cracks

Before destruction, two dangerous inclined cracks appeared on both sides of the middle support in the research spans; as a rule, the opening widths of these were much greater than others and were wedged into concrete compressed zones above the others. Sometimes, immediately before destruction, two dangerous unifying inclined cracks formed from several closely spaced inclined cracks.

The length of dangerous inclined cracks, brought to the working height of the cross-section $h_0$, destroyed over the middle supports of continuous reinforced concrete beams, can be determined by the model:

$$\hat{Y}(c_0/h_0)_V = 1.37 + 0.56x_1 - 0.06x_2 - 0.07x_3 - 0.21x_4 + 0.17x_5 - 0.05x_5^2 - 0.05x_1x_2 - \\ -0.07x_1x_3 - 0.15x_1x_4 + 0.12x_1x_5 - 0.05x_3x_4 + 0.05x_3x_5 - 0.03x_4x_5, \qquad v = 6\%. \tag{8}$$

Analysis of the studies showed that, in continuous beams, the average projection length of dangerous inclined cracks exceeded 24% of that for ordinary single-span elements.

*2.6. Transverse Force Perceived by Longitudinal Reinforcement*

Processing the data of pairwise located strain gauges glued to the longitudinal lower and upper reinforcement in adjacent sections of the slice spans (according to the rules of construction theory) allowed us to determine the increase in bending moments in these sections. Through them, we also determined the values of the maximum transverse forces ($V_{sH}$ and $V_{sb}$) that are perceived—accordingly, the lower and upper longitudinal reinforcement in the supporting sections of the samples. Thus, the transverse force, reduced to $V_u$ destructive, was perceived by the lower and upper longitudinal reinforcement of the beam, respectively:

$$\hat{Y}(V_{Sb}/V_u) = 3.0 - 0.6x_1 - 0.3x_2 + 1.0x_4 - 0.4x_5 - 0.5x_1^2 + 0.8x_1x_2 + \\ +0.3x_1x_3 - 0.6x_1x_4 - 0.5x_3x_4, \%, \qquad v = 6\%; \tag{9}$$

$$\hat{Y}(V_{st}/V_u) = 1.8 - 0.3x_1 - 0.2x_2 + 0.07x_5 - 0.2x_1^2 - 0.2x_2x_4 - \\ -0.3x_2x_5 + 0.4x_1x_3 - 0.4x_3x_5 - 0.3x_1x_5, \%, \qquad v = 6\%. \tag{10}$$

The analysis of relationships (9) and (10) showed, first, a relatively small average value of the transverse force in the longitudinal lower and upper reinforcement (3.0% and 1.8%, respectively), which intersected with dangerous inclined sections above the average support and under concentrated forces. Second, we noted that, with an increase in the diameters of the longitudinal working reinforcement and a decrease in the number of transverse reinforcement, part of the transverse force perceived by the longitudinal reinforcement grew insignificantly. The low value of $V_{su}$ in the longitudinal reinforcement, compared with published data by other authors, can be explained; first, the research elements are nonremedial (i.e., ($\rho_{lb}$ and $\rho_{lt}$) are in the optimal region), and second, effective operation of a sufficiently strong transverse reinforcement takes up a significant part of the transverse force and does not allow free bending of the longitudinal reinforcement, which is crossed by a dangerous inclined crack.

Therefore, the destruction of continuous reinforced concrete beams and tape grillages above the average support or pile, under the combined action of transverse forces and bending moments caused by transverse concentrated forces F or distributed load, occurs (Figure 4) in the form of an inverted trapezoid with the possible formation of so-called plastic hinges over the middle support and under concentrated forces, as well as the possible redistribution of internal forces.

**3. Comparison of Experimental and Calculated Values of the Strength of the Supporting Sections of the Experimental Elements, according to the Recommendations of National Design Standards**

In recent years, there has been has renewed interest in the study of issues related to the strength of inclined and supporting sections of reinforced concrete structures. The journal *ASI* has published more than 1000 articles on the issues of transverse destruction of reinforced concrete structures. However, the calculation methods that have been introduced into the design standards of most countries of the world remain conservative. For example, if EUROCODE-2, domestic DBN B.2.6-98, and DSTU B.V.2.6-156: 2010 made significant adjustments to many sections of the rules, then the provisions relating to the calculation of oblique sections of structures that remain at the models' positions form an analogy with a variable angle of inclination of the concrete strut.

The German standards (DIN1045-1.12: 2008) contain a calculation model based on the provisions of the modified truss analogy, which considers the shear component perceived by the inclined section due to the adhesion forces on the banks of the diagonal crack. However, when calculating the transverse reinforcement, the angle of inclination of the compressed concrete strut in them is recommended to be assumed a constant ($\theta = 40°$).

Studies have shown that the improvement of calculation methods based on the model of truss analogy have led to an increasing number of calculation formulas of empirical origin. For example, ACICODE 318-08R norms included 431 formulas that took into account different loading conditions and were used to design individual types of elements of the same design.

One of the objectives of this study was to verify the reliability and safety of the strength forecast for supporting sections of continuous reinforced concrete beams and high grillages according to the most common foreign regulatory documents. The experimental and calculated values of the destructive shear force, determined according to the recommendations of these standards, are given in [16,34].

An analysis of the results shows that the previously applicable SNiP 2.03.01-84 * and the new Russian standards have better convergence (the coefficients of variation $\upsilon$ are equal to 27.3% and 43.3%), respectively, of the calculated and experimental data of the bearing capacity of the supporting sections of the experimental beams in comparison with other regulatory documents [16,34] because they are better than other foreign standards and reflect the physical picture of the work.

### 3.1. Modeling the Complex Stress-Strain State of Research Elements

The stress-strain state of the research elements was modeled in a nonlinear setting using the «LIRA-Sapr» finite element PC, which is based on the general theory of reinforced concrete with cracks developed by N.I. Karpenko and his students. Additionally, we made use of full and two-line diagrams of deformation, respectively, of concrete and reinforcement. The tensile strength of concrete with a complex heterogeneous stress state of the test samples was determined automatically using the phenomenological strength criterion of G.A. Geniev, V.M. Kissyuk, and G.A. Tyupin, embedded in the specified software package. For modeling in Lira-Sapr PC, the tensile strengths of concrete from Table 1 were indicated.

**Table 1.** Nominal physical and mechanical characteristics of materials.

| N/N | Characteristics of the Material | Concrete at the Age of 28 Days | | | Type of Reinforcement | |
|---|---|---|---|---|---|---|
| | | C12/15 | C20/25 | C30/35 | A500C | Вр1 |
| 1 | Maximum compressive strength, MPa | 15.3 | 25.5 | 35.7 | 500 | 395 |
| 2 | Maximum tensile strength, MPa | 1.69 | 2.337 | 2.97 | 500 | 395 |
| 3 | Initial modulus of elasticity, $E_{ck}$, MPa | 23,665 | 30,600 | 34,995 | 190,000 | 200,000 |
| 4 | Compressibility, $\varepsilon_{cl}$ $10^{-5}$, $\varepsilon_{s0}$ $10^{-5}$ | 166 | 181 | 194 | 263 | 198 |
| 5 | Extreme compressibility, $\varepsilon_{cu}$ $10^{-5}$, $\varepsilon_{su}$ $10^{-5}$ | 496 | 395 | 321 | 2500 | 2500 |
| 6 | Extreme extensibility, $\varepsilon_{ctu}$ $10^{-5}$, $\varepsilon_{stu}$ $10^{-5}$ | 16.6 | 18.1 | 19.4 | 2500 | 2500 |
| 7 | Poisson's ratio, $\nu_c$ | 0.2 | 0.2 | 0.2 | 0.25 | 0.25 |

Based on the symmetry of the research elements, the calculation was carried out only for the left half of the beam. The results of the numerical implementation of full-scale experiments with continuous reinforced concrete beam-grillage are presented in Figure 6 and Table 2.

Verification of the obtained data was carried out by comparison with other methods [19–31,38].

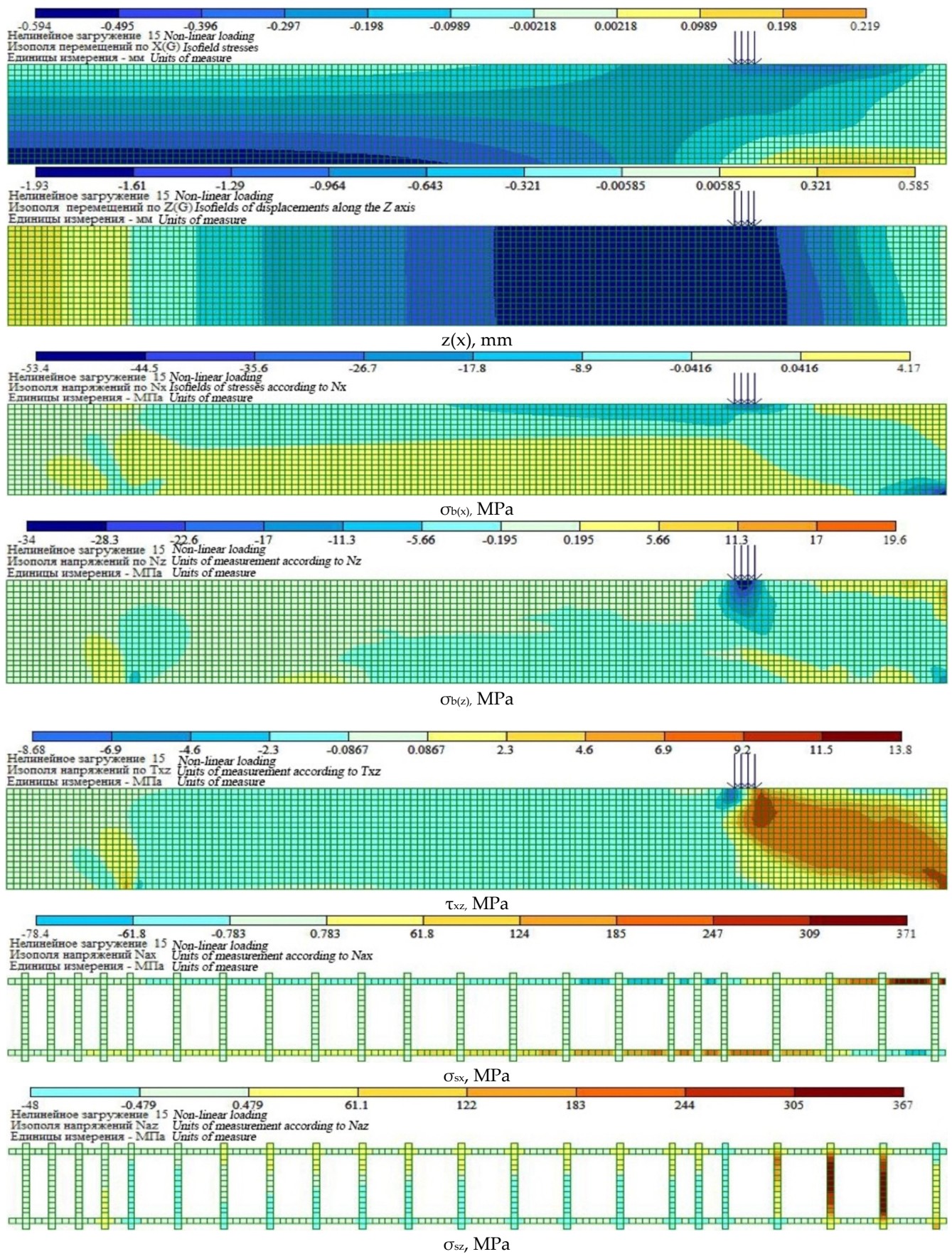

**Figure 6.** Isofield of displacements and stresses of a continuous beam destroyed according to the punching scheme on the middle support (*F/V* scheme), experiment No. 27.

**Table 2.** The results of modeling the SSS of the supporting sections of continuous reinforced concrete beams (series *V*) before the destruction ($F \cong 0.95F_u$) of their supporting sections, according to the bursting pattern above the middle support (*F/V*). Experience No. Deflection, $-z$, mm Stress in concrete and reinforcement, MPa.

| No. Experience | Deflection, $-z$, [mm] | | Stress in Concrete and Reinforcement [MPa] | | | | | | | | | |
| | | | $-\sigma_{cx}$ | | $-\sigma_{cz}$ | | $+\tau_{sxz}$ | $\sigma_{cx,b}$ | | $\sigma_{cx,t}$ | | $\sigma_{swz}$ |
| | $x = l/2$ | $x = l-a$ | $x = l-a$ | $x = l$ | $x = l-a$ | $x = l$ | $x = l-a...l$ | $x = l-a$ | $x = l$ | $x = l-a$ | $x = l$ | $x = l-a...l$ |
|---|---|---|---|---|---|---|---|---|---|---|---|---|
| 1 | 1.57 | 1.57 | 24.8 | 39.6 | 25.2 | 19.8 | 14.3 | 265 | −138 | −99 | 398 | 68 |
| 2 | 3.31 | 1.40 | 25.2 | 29.4 | 25.1 | 25.2 | 12.2 | 206 | +163 | −41 | 247 | 393 |
| 3 | 0.56 | 0.47 | 9.1 | 27.2 | 38 | 20.7 | 18.1 | 122 | −41 | −1 | 365 | 79 |
| 4 | 1.80 | 1.80 | 25.3 | 30.3 | 20.4 | 17.0 | 8.3 | 286 | −143 | −71 | 429 | 155 |
| 5 | 0.26 | 1.39 | 22.7 | 34.1 | 40.4 | 26.9 | 12.5 | 122 | +31 | +31 | 367 | 402 |
| 6 | 2.81 | 3.75 | 15.5 | 24.8 | 24.8 | 13.5 | 8.4 | 304 | −76 | −76 | 304 | 402 |
| 7 | 1.62 | 1.62 | 28.9 | 43.4 | 23.5 | 15.7 | 12.9 | 286 | −124 | −82 | 491 | 32 |
| 8 | 0.63 | 0.84 | 10.0 | 17.4 | 22.4 | 14.9 | 10.5 | 96 | +48 | −36 | 288 | 161 |
| 9 | 0.91 | 1.09 | 10.9 | 32.7 | 36.9 | 24.6 | 19.8 | 82 | +41 | −1 | 491 | 215 |
| 10 | 2.79 | 2.09 | 17.4 | 26.0 | 21.1 | 15.3 | 9.7 | 245 | −82 | −163 | 491 | 198 |
| 11 | 1.67 | 1.67 | 29.3 | 39.0 | 25.8 | 14.3 | 12.8 | 263 | −175 | −66 | 395 | 72 |
| 12 | 2.38 | 4.08 | 21.6 | 25.9 | 22.4 | 22.4 | 12.8 | 321 | +214 | −1 | 241 | 402 |
| 13 | 0.90 | 1.20 | 17.1 | 22.8 | 38.4 | 25.6 | 20.3 | 154 | +31 | −22 | 369 | 287 |
| 14 | 0.99 | 0.99 | 22.2 | 25.4 | 20.5 | 12.3 | 9.1 | 391 | −231 | −65 | 326 | 239 |
| 15 | 1.52 | 1.52 | 29.0 | 33.8 | 27.2 | 8.2 | 9.6 | 163 | −82 | −82 | 491 | 32 |
| 16 | 1.30 | 1.95 | 13.9 | 17.4 | 18.3 | 13.5 | 10.0 | 107 | +107 | −27 | 320 | 402 |
| 17 | 2.22 | 2.22 | 21.1 | 25.3 | 19.7 | 16.9 | 10.4 | 286 | −122 | −82 | 491 | 138 |
| 18 | 0.29 | 1.55 | 13.6 | 18.1 | 23.2 | 16.6 | 11.6 | 129 | +64 | −26 | 386 | 398 |
| 19 | 1.58 | 1.58 | 17.1 | 28.5 | 27.6 | 17.3 | 16.8 | 163 | −101 | −82 | 491 | 174 |
| 20 | 1.83 | 2.44 | 15.6 | 15.6 | 17.6 | 11.7 | 9.1 | 198 | −29 | −56 | 339 | 365 |
| 21 | 1.71 | 1.71 | 17.5 | 26.3 | 24.6 | 16.4 | 11.7 | 185 | −74 | −74 | 443 | 200 |
| 22 | 1.37 | 1.37 | 20.4 | 24.5 | 23.0 | 15.3 | 11.6 | 162 | −65 | −65 | 388 | 189 |
| 23 | 1.66 | 2.22 | 18.6 | 23.3 | 23.4 | 17.5 | 12.5 | 152 | −39 | −76 | 457 | 365 |
| 24 | 1.20 | 1.44 | 16.7 | 25.1 | 23.0 | 15.3 | 12.0 | 241 | −108 | −69 | 413 | 182 |
| 25 | 1.45 | 1.93 | 17.8 | 26.7 | 22.6 | 17.0 | 12.7 | 216 | −78 | −62 | 371 | 336 |
| 26 | 1.28 | 1.28 | 16.3 | 24.4 | 22.2 | 17.3 | 12.2 | 199 | −40 | −80 | 478 | 129 |
| 27 | 1.11 | 1.33 | 16.3 | 24.5 | 22.4 | 15.3 | 11.7 | 196 | −79 | −65 | 393 | 171 |
| Average values | 1.51 | 1.72 | 19.0 | 27.1 | 25.2 | 17.3 | 12.4 | 205 | −42 | −57 | 395 | 229 |

An analysis of the results of modeling the stress-strain state of reinforced concrete elements showed that the use of a nonlinear finished-element calculation, based on the general mechanics of reinforced concrete using phenomenological strength criteria, made it possible to fully reproduce the results of a full-scale experiment. In particular, their fracture, according to the punching scheme over medium support from the overwhelming effect of cracking shear stresses in concrete $\tau_{cxz} \approx 0.5_{fck}$, was reproduced with sufficient accuracy for practical x calculations ($v = 9\%$).

Consistent analysis of the Isopole stresses, displacements, and forces in materials used for construction allows one to numerically confirm the influence of research structural factors and external factors on the bearing capacity of continuous reinforced concrete beams and grillages. Additionally, such analyses can predict the nature of further deformations and fractures.

### 3.2. Engineering Method for Calculating the Strength of the Supporting Sections of Continuous Reinforced Concrete Beam-Grillage

Despite the increase in the number of publications predicting the strength of inclined sections of reinforced concrete structures, the calculation methods that have been introduced into the design standards of most countries remain conservative and based on truss analogies. The information given above showed that, in the calculations of the bearing capacity of the supporting sections of continuous reinforced concrete beams and grillages, it is necessary to move away from stereotypes and implement calculations and designs according to the forcing pattern (**F/V**) over the middle support. The nature of the destruction of all prototype beams was confirmed by clear photo fixation, as given in monograph [34], and by the results of the above simulation of their stress-strain state.

The fracture scheme of continuous reinforced concrete grillage beams took the form of an inverted trapezoid (Figure 7) with significant shear stresses (up to $0.5 f_{ck}$) on the lateral faces of the so-called punching shear pyramid, relatively small average tensile stresses in transverse reinforcement (at the level of $0.6 f_{ywk}$) and intersected by inclined cracks), as well as relatively small stresses in the lower and upper longitudinal reinforcement in the places where these cracks intersected.

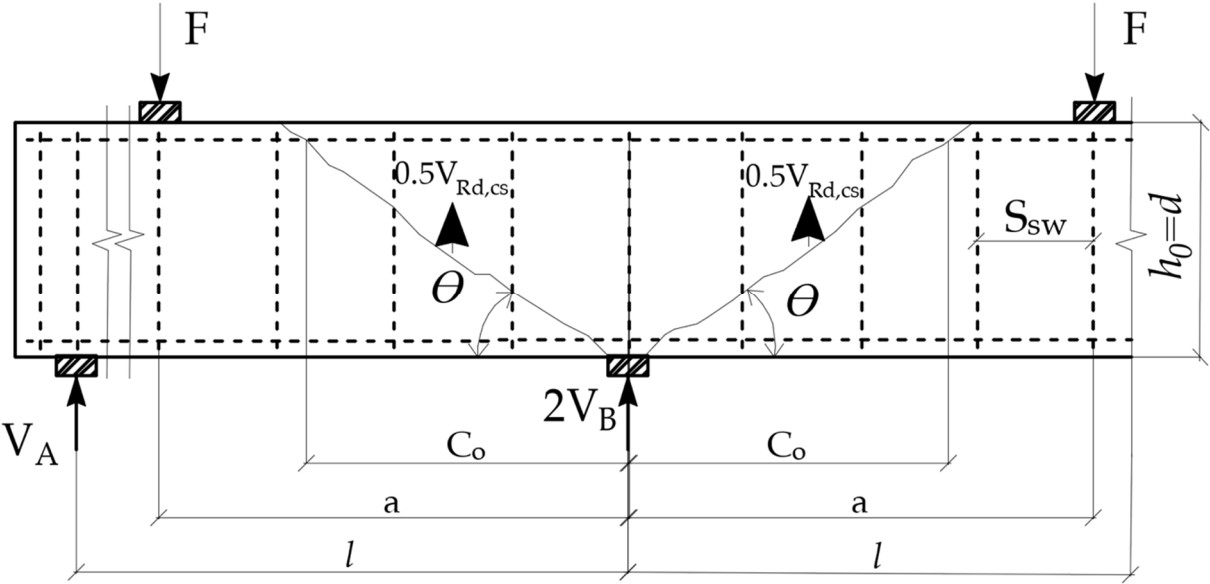

**Figure 7.** Diagram of the destruction of continuous reinforced concrete beams.

At the same time, the bursting body, as a rule, was divided by one normal (or several filament) fissure into separate blocks with the possible formation of conditional plastic joints over the middle support (more often) and in the spans (less often).

Comprehensive analysis of the obtained experimental data showed that the punching body of the continuous beam and grillage above the middle support qualitatively coincided with the punching shear pyramid, according to previously effective domestic norms [10], with a variable angle of inclination of the side faces. Based on the above, and taking into account the results of field experiments and modeling data, calculation of continuous reinforced concrete beam-grillages for punching over the middle support B can be determined by the condition:

$$F_B = 2V_B \leq F_c + F_{sw} = \alpha_c f_{ctd} U_m h_0 + \beta_w f_{ywd} \sum A_{sw.2C_0}, \tag{11}$$

where: $\boldsymbol{F_B}$—is the reaction value of the middle support;

- $2\,\boldsymbol{V_B}$—calculated values of transverse forces, respectively, to the left and the right of the support;

- — $F_c$—cutting efforts, perceived by concrete as the lateral faces of the punching body;
- — $F_{sw}$—cutting efforts, perceived by transverse reinforcement on the lateral faces of the punching body with the total area $\sum A_{sw,2C_0}$;
- — $\alpha = \sigma_c / f_{ctk}$—a coefficient characterizing the level of vertical stresses in concrete on the lateral faces of the conditional bursting pyramid as determined by (12);
- — $\beta_w = \sigma_{sw}/f_{yk}$—coefficient, considering the level of stresses in the transverse reinforcement of the supporting sections, which is intersected by the inclined side faces of the punching body and is determined by (13);
- — $U_m$—is the arithmetic mean of the perimeters of the upper and lower base of the body of the punching within the working height of the section $h_0 = d$.

Empirical expressions for the coefficients $\alpha_c$ and $\beta_w$ were obtained from the corresponding adequate mathematical models with good information content by replacing the coded values of design factors with natural values:

$$
\begin{aligned}
\alpha_s =\ & 1.65 - 0.94(a/h_0 - 2) - 0.13\left(\tfrac{s-25MRa}{10MRa}\right) - 0.15\left(\tfrac{\rho_w - 0.0034}{0.0016}\right) + 0.29 \times \left(\tfrac{\rho_{l.t}-0.015}{0.005}\right) + \\
& + 0.26\left(\tfrac{\rho_{l.b}-0.015}{0.005}\right) - 0.49(a/h_0-2)^2 + 0.14(a/h_0-2)\times\left(\tfrac{c-25MRa}{10MRa}\right) - 0.31(a/h_0-2)\times \\
& \times\left(\tfrac{\rho_{l.t}-0.015}{0.005}\right) - 0.006\times\left(\tfrac{s-25MRa}{10MRa}\right)\left(\tfrac{\rho_w-0.0034}{0.0016}\right) - 0.05\left(\tfrac{s-25MRa}{10MRa}\right)\times\left(\tfrac{\rho_{l.b}-0.015}{0.005}\right) + \\
& + 0.07\times\left(\tfrac{\rho_w-0.0034}{0.0016}\right)\times\left(\tfrac{\rho_{l.t}-0.015}{0.005}\right), v = 7.9\%;
\end{aligned}
\tag{12}
$$

$$
\begin{aligned}
\beta_w =\ & 0.61 - 0.17(a/h_0-2) - 0.17\left(\tfrac{s-25MRa}{10MRa}\right) - 0.03\left(\tfrac{\rho_w-0,0034}{0.0016}\right) + 0.10\times \\
& \times\left(\tfrac{\rho_{l.t}-0.015}{0.005}\right) + 0.15\left(\tfrac{\rho_{l.t}-0.015}{0.005}\right) - 0.04\left(\tfrac{\rho_{l.t}-0.015}{0.005}\right)^2 + 0.03(a/h_0-2)\times \\
& \times\left(\tfrac{\rho_{l.t}-0.015MRa}{0.005MRa}\right) - 0.03(a/h_0-2)\left(\tfrac{\rho_{l.t}-0.015}{0.005}\right) + 0.006\left(\tfrac{s-25MRa}{10MRa}\right)\times\left(\tfrac{\rho_w-0.0034}{0.0016}\right) - \\
& - 0.06\left(\tfrac{\rho_w-0.0034}{0.0016}\right)\times\left(\tfrac{\rho_{l.t}-0.015}{0.005}\right) - 0.03\times\left(\tfrac{\rho_w-0.0034}{0.0016}\right)\times\left(\tfrac{\rho_{l.t}-0.015}{0.005}\right), v = 6.8\%.
\end{aligned}
\tag{13}
$$

For beams of rectangular section, expression (11) takes the form:

$$
F_v = F_c + F_{sw} = \alpha_c f_{ctd} 2 b h_0 + \beta_w f_{ywd}\{[c + 2(s_0/h_0)h_0]/S_w\}A_{sw},
\tag{14}
$$

where $c$ is the width of the area (pile caps) of the transmission of the support reaction;

$A_{sw}$ is the cross-sectional area of the transverse rods in one transverse plane;

$c_0/h_0$ is the relative length of the horizontal projection of the dangerous inclined crack, which is recommended to be (15) obtained from the mathematical model (8) by replacing the coded values of research factors with their natural values:

$$
\begin{aligned}
c/h_0 =\ & 1.37 + 0.56(a/h_0-2) - 0.06\left(\tfrac{s-25MRa}{10MRa}\right) - 0.07\left(\tfrac{\rho_w-0.0034}{0.0016}\right) - 0.21\times\left(\tfrac{\rho_{l.b}-0.015}{0.005}\right) + \\
& + 0.17\left(\tfrac{\rho_{l.t}-0.015}{0.005}\right) - 0.05\left(\tfrac{\rho_{l.t}-0.015}{0.005}\right)^2 - 0.05(a/h_0-2)\times\left(\tfrac{c-25MRa}{10MRa}\right) - 0.07(a/h_0-2)\times \\
& \times\left(\tfrac{\rho_w-0.0034}{0.0016}\right) - 0.15(a/h_0-2)\left(\tfrac{\rho_{l.b}-0.015}{0.005}\right) + 0.12(a/h_0-2)\left(\tfrac{\rho_{l.t}-0.015}{0.005}\right) - 0.05\left(\tfrac{\rho_w-0.0034}{0.0016}\right)\times \\
& \times\left(\tfrac{\rho_{l.b}-0.015}{0.005}\right) + 0.05\left(\tfrac{\rho_w-0.0034}{0.0016}\right)\left(\tfrac{\rho_{l.t}-0.015}{0.005}\right) - 0.03\left(\tfrac{\rho_{l.b}-0.015}{0.005}\right)\left(\tfrac{\rho_{l.t}-0.015}{0.005}\right), v = 6\%.
\end{aligned}
\tag{15}
$$

When determining the normative value of the forcing force $F_b$ according to Equation (9) and (12), $f_{ctk}$ and $f_{ywk}$ should be taken instead of $f_{ctd}$ and $f_{ywd}$.

Comparison of the experimental values and those calculated by the proposed engineering method of the bearing capacity of the experimental continuous beams/grillages for punching over the middle support showed satisfactory convergence ($v \le 9\%$).

## 4. Conclusions

The complex experimental and theoretical studies carried out solved the urgent scientific and technical problem of calculating the bearing capacity near the support sections for punching continuous reinforced concrete beams and high grillages.

1.  Thanks to the adopted methodology, new experimental data were obtained and physical models of continuous reinforced concrete structures were substantially refined. As a result, the systemic influence on the deformability, fracture toughness, and strength of the prototype beams for the relative shear span $a/h_0$, concrete class $C$, transverse coefficient $\rho_w$, working $\rho_{l,b}$, and mounting $\rho_{l,t}$ are reinforced, both in particular and during their interactions. The obtained adequate experimental and statistical dependences (mathematical models) of the calculated parameters of the bearing capacity of the research elements made it possible to comprehensively solve optimization problems.

2.  The features of the stress-strain state of the experimental beams were disclosed. The well-known features were systematized and a new scheme ($F/V$) for the destruction of continuous reinforced concrete beam-grillages was discovered. The formation of conditional plastic joints was revealed due to the nonlinearity of deformation of their materials.

3.  An analysis of methods used for calculating the strength of supporting sections of spans of reinforced concrete structures as set down in national design standards in developed countries, as well as the author's methods, showed that the vast majority of these standards were based, not on new general methods, but on partially improved methods used in the past. In particular, the calculation methods of ES-2 and other foreign countries are based on various conventional schemes and analogies.

4.  Comparison of the calculated and experimental values of the bearing capacity of continuous reinforced concrete structures (calculated according to the recommendations of various national design standards) showed unsatisfactory convergence and insufficient reliability of the calculation formulas. For some prototypes, including those with large shear spans ($a/h_0 = 3$), the design strength significantly exceeded their actual bearing capacity.

5.  Modeling the complex stress-strain state of experimental structures by non-linear finite element calculations using the tested «LIRA-Sapr» software package made it possible to follow all stages of their work under load and numerically reproduce the results of the performed experiments, making a reliable prediction of their strength and the nature of their failure.

6.  The proposed engineering method for calculating the strength near the supporting sections of continuous reinforced concrete beams and high grillages makes it possible to make a reliable forecast of their bearing capacity with satisfactory accuracy for practical purposes (coefficient of variation $v \leq 9\%$).

**Author Contributions:** Conceptualization, Z.K. and I.K.; methodology, Y.K., I.G. and I.K.; validation, Z.K., Z.K. and I.K.; formal analysis, Y.K. and I.K.; resources, Z.K. and I.G.; data curation, I.K. and Y.K.; writing—original draft preparation, Z.K., I.K. and I.G.; writing—review and editing, Z.K. and I.G.; visualization, I.K. and Y.K.; supervision, Z.K. and I.K.; project administration, I.K. and Y.K.; funding acquisition, Z.K. All authors have read and agreed to the published version of the manuscript.

**Funding:** This research received no external funding.

**Institutional Review Board Statement:** Not applicable.

**Informed Consent Statement:** Not applicable.

**Data Availability Statement:** The results of the experiment used to build the design model are published in monograph [16,34].

**Conflicts of Interest:** The authors declare no conflict of interest.

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
