# Peer review of "Bearing Capacity near Support Areas of Continuous Reinforced Concrete Beams and High Grillages"

_applsci, doi:10.3390/app12020685_

Round 1

Reviewer 1 Report

The manuscript needs a thorough linguistic revision. I suggest to send it to a native speaker.

Chapter 1. 2 should use different wording, for example:

“Select...” should be “Selection...”;

“Assess...” should be “Assessment...”;

“Develop...” should be “Development...”.

Chapter 2 begins with “In this regard…” – what does this beginning of the sentence refer to? Maybe it would be better to skip it? It looks like a continuation of earlier sentences - that's not how you should start writing chapters.

Authors wrote: „To achieve this goal, a series of natural experiments with two-span continuous reinforced concrete beams were implemented on the relevant state budget topics…” – instead of “natural experiments” it would be more appropriate to use the phrase: “field tests” or “in-situ experiments/investigations” or “natural scale research”.

Take a look there, please:

https://en.wikipedia.org/wiki/Natural_experiment: “A natural experiment is an empirical study in which individuals (or clusters of individuals) are exposed to the experimental and control conditions that are determined by nature or by other factors outside the control of the investigators. The process governing the exposures arguably resembles random assignment.”

Figure 1 is illegible. The descriptions are of different sizes, plus the letters and numbers are distorted. The cross-sections of reinforced concrete beams do not show the location of supports! Figure 2 also does not indicate where exactly the supports are located.

In Figure 2, the measurement sensor numbers are described in "Cyrillic" - there are other designations in the text of the manuscript.

Figures 3 and 4 are also of poor quality, and some of the descriptions are not in English.

There is no photographs in the manuscript documenting the experimental studies conducted. Only fragments of theoretical calculations are presented. It is not known if the research was done at all.

Reviewer 2 Report

1 L335.L372.L379. If possible, please validate these equations on other samples in previous research.
2 If possible, please add photos of the experimental process. Meanwhile, the broken samples should be shown.
3 L225. Please illustrate the materials in detail.
4 L483. "The tensile strength of concrete with a complex heterogeneous stress state of the test samples was determined automatically using the phenomenological strength criterion of G.A. Geniev, V.M. Kissyuk, and G.A. Tyupin, embedded in the specified software package." Different types of concrete have totally different tensile strengths. Please add the citations and clarify the specific tensile strength.
Ref.:
Influence of High Temperature Curing and Surface Humidity on the Tensile Strength of UHPC. MATERIALS, 2021,14.
Review of Cementitious Composites Containing Polyethylene Fibers as Repairing Materials. Polymers 2020, 12, 2624.
A two-dimensional micromechanical damage-healing model for microcapsule-enabled self-healing cementitious composites under tensile loading. International Journal of Damage Mechanics, 2015, 24(1): 95–115. 

Round 2

Reviewer 1 Report

Dear Authors,

It is not my intention to negate the publication of the article. I just wish it was more accessible to the reader, hence my criticisms.

The content in terms of substantive is valuable. The conclusions are supported by the obtained research results and can be used in future engineering practice.

Unfortunately the drawings are made carelessly (Figure 1 attached with irregularities marked).

Dimension descriptions touch or intersect with dimension lines.

The dimension lines are at different distances from each other.

The other drawings are done in a similar style.

The added new figures sufficiently document the execution of the experimental tests.
